# Establishment of Critical Nitrogen Concentration Models in Winter Wheat under Different Irrigation Levels

**Bin-Bin Guo [1,2], Xiao-Hui Zhao [1], Yu Meng [1], Meng-Ran Liu [1], Jian-Zhao Duan [1], Li He [1], Nian-Yuan Jiao [2], Wei Feng [1,*,†] and Yun-Ji Zhu [1,*]**

[1] National Engineering Research Centre for Wheat, State Key Laboratory of Wheat and Maize Crop Science, Henan Agricultural University, #15 Longzihu College District, Zhengzhou 450046, China; Guobin90@126.com (B.-B.G.); 15736705088@stu.henau.edu.cn (X.-H.Z.); 15736749779@stu.henau.edu.cn (Y.M.); L18703825181@stu.henau.edu.cn (M.-R.L.); djz20008@163.com (J.-Z.D.); he-li19870308@163.com (L.H.)

[2] College of Agronomy, Henan University of Science and Technology, Luoyang 471023, China; jiaony1@163.com

[*] Correspondence: fengwei78@126.com (W.F.); hnndzyj@126.com (Y.-J.Z.)

[†] Current address: College of Agronomy, Henan Agricultural University, #15 Longzihu College District, Zhengzhou 450046, China.

**Abstract:** The aim of this study was to verify the applicability of the critical nitrogen concentration dilution curve (Nc) of wheat grown under different irrigation conditions in the field, and discuss the feasibility of using the N nutrition index (NNI) to optimize N fertilizer application. The high-yield, medium-protein wheat varieties Zhoumai 27 and Zhoumai 22 were used in field experiments in two different locations (Zhengzhou and Shangshui) in Huang-Huai, China. Plants were grown under rainfed and irrigation conditions, with five N application rates. Nc models of the leaves, stems, and whole plant were constructed, followed by establishment of an NNI model and accumulative N deficit model ($N_{and}$). As previous research reported, our results also showed that the critical N concentration and biomass formed a power function relationship ($N = aDW^{-b}$). When the biomass was the same, the critical N concentration was higher under irrigation than rainfed treatment. Meanwhile, the fitting accuracy ($R^2$) of the Nc model was also higher under irrigation than rainfed treatment in both sites, and was higher in the stems and whole plant. The NNI calculated using the Nc model increased with increasing N application, reflecting N deficiency. Moreover, there was a significant negative linear correlation between NNI and $N_{and}$, and both indices could be uniformly modeled between locations and water treatments. The accuracy of the $N_{and}$ model was highest in the whole plant, followed by the leaves and stems. The models constructed in this paper provide a theoretical basis for accurate management of N fertilizer application in wheat production.

**Keywords:** wheat; water conditions; critical nitrogen concentration; nitrogen nutrition index; accumulative nitrogen deficit

## 1. Introduction

Henan Province is the main wheat producing area in China, representing 22.1% of the entire production area and producing 25% of the national wheat output. N is the most demanding element of wheat growth, and wheat is particularly sensitive to N levels in soils. Insufficient N fertilizer limits crop growth, while excessive application leads to a low N utilization rate, thereby wasting resources and increasing environmental pollution. Optimal N application is therefore a key factor regulating

wheat growth and development, improving photosynthetic performance and increasing yield and quality [1,2]. Traditional methods of N nutrition diagnosis include analyses of growth, leaf color, and overall symptoms and so on. However, these diagnostic tools tend to rely on experience and subjective judgment, and, therefore, the likelihood of error is relatively large, greatly restricting their accuracy and application. Meanwhile, although chemical diagnostic methods are more accurate, they are both time and labor consuming. In addition, methods aimed at determining the N content of a plant differ between different growth stages and under different growing conditions. Accurate diagnosis of the N status at various stages of growth is therefore vital for the optimal management of N fertilizer application in crop fields.

The "critical nitrogen concentration", that is, the minimum N concentration value required to obtain maximum biomass growth, is often recommended as a diagnostic tool [3,4]. Determining the critical nitrogen concentration of a crop is therefore one method of nitrogen nutrition diagnosis, and is usually based on the nitrogen concentration of crop-specific components such as the leaves, stems, or whole plant. Studies have shown that the N concentration in crops decreases with increasing biomass, while the relationship between the two can be expressed by the power function equation $N_C = aDW^{-b}$ (Nc: critical N concentration value (%), DM: maximum dry matter accumulation (t hm$^{-2}$), a: critical N concentration value when the dry matter is 1 t hm$^{-2}$, and b: the slope of the curve), which represents the critical N concentration dilution curve. The N content of wheat undergoes a dilution process with increasing dry matter. This dilution phenomenon is mainly caused by two processes, the mutual shading of leaves and the change in ratio of leaf to stem dry matter with growth. In line with this, Lemaire and Salette (1984) [5] suggested that there is a critical value for the N content of aboveground dry matter during crop growth and development; namely, the critical N concentration. They hypothesized that when the actual N content of a crop is lower than the critical concentration, crop growth will be limited. Meanwhile, concentrations above this critical value indicate excessive N, while those close to the critical value suggest optimal N for crop growth. The critical N concentration value is therefore a very important indicator of crop growth and N nutrition diagnosis.

Greenwood et al. (1990) [3] proposed a general model for critical N concentrations and aboveground dry matter in C3 and C4 crops; namely, $N = 5.7 DW^{-0.5}$ and $N = 4.1 DW^{-0.5}$, respectively, which Lemaire et al. (1990) [6] subsequently revised to $N = 4.8 DW^{-0.34}$ and $N = 3.6 DW^{-0.346}$. However, these models were based on multiple test means, and do not necessarily represent all C3 and C4 crops. More recently, parameter models were, therefore, established for multiple crops, including maize [4], rice [7], oilseed rape [8], sorghum [9], and tomato [10]. For example, Justes et al. (1994) [11] constructed a dilution curve model for the aboveground critical N concentration of winter wheat in France ($N = 5.35DW^{-0.44}$), while Ziadi et al. (2010) [12] constructed an aboveground N dilution model for spring wheat in Canada ($N = 3.85DW^{-0.57}$). In China, Zhao et al. (2012) [13] and Yue et al. (2016) [14] constructed dilution curve models of the aboveground critical N concentration of winter wheat in Eastern China ($N = 4.42DW^{-0.44}$) and the North China Plain ($N = 4.15DW^{-0.38}$), respectively. Parameters a and b differ in all of the above models, probably due to differences in the test crops and ecological conditions. Understanding the applicability and reliability of models for a specific crop in different ecological regions and under different cultivation conditions is therefore important.

Water is an important raw material for photosynthesis and an important factor affecting crop growth in farmland ecosystems, especially wheat producing areas in arid and semi-arid regions, where the effect of drought on growth and yield is significantly greater than all other environmental factors. Furthermore, the effects of water and fertilizer application on crop growth are not isolated, but instead interact with each other [15–17]. For example, studies have shown a significant positive correlation between N fertilizer application and irrigation under normal irrigation conditions. Moreover, as soil drought gradually deepens, the positive relationship between water and fertilizer tends to decrease or become less obvious [18,19]. However, if the soil water is severely deficient, the application of N fertilizer has a negative effect [20]. The lack of water limited fertilizer efficiency, while excessive water led to fertilizer leaching and yield reductions [21,22]. Optimal irrigation therefore improves the

availability of soil nutrients, promoting the absorption and utilization of soil nutrients by crop roots and increasing the transfer of nutrients to crop grains, thereby increasing overall yield [23]. Coelho and Or (1998) [24] revealed that water deficits cause an increase in the proportion of roots in deep soil, benefiting the absorption and utilization of water and nutrients, and thereby stabilizing crop yield. Meanwhile, Li et al. (2009) [17] showed that reasonable nitrogen supply under water stress resulted in an increase in nitrate reductase activity and a higher protein content in wheat leaves. In crop production, irrigation should therefore be combined with fertilization, helping regulate fertilizer use, while optimizing N application, and improving overall efficiency. However, most critical N concentration dilution models were constructed under sufficient irrigation conditions. Under water-limited conditions, the N uptake and transport characteristics of plants change greatly, and, thus, the N dilution model of crops under sufficient irrigation conditions are no longer applicable.

Studies on critical N models under different irrigation conditions are currently limited to horticultural crops such as tomatoes [10] and sweet peppers [25], with few studies having determined the critical N concentration of open-field food crops under different irrigation levels. In this study, we conducted field experiments under different N fertilizer gradients and different irrigation conditions in two ecological regions in Henan Province, China. The aims were to determine the critical N concentration dilution curves of different wheat parts and clarify the applicability of these models in wheat-producing areas of Huang-Huai-Hai Plain. We also estimated N deficits, and the feasibility of using the N nutrition index (NNI) to optimize N fertilizer application in wheat. The findings provide a theoretical basis for accurate management of N fertilizer application in wheat production.

## 2. Materials and Methods

### 2.1. Experiment Design

The experiments were carried out at Henan Province, China (Zhengzhou: 34°51′ N, 113°35′ E and Shangshui: 33°33′ N, 114°37′ E) over a 3-year period. The different experimental conditions, N fertilizer levels, irrigation regimes, wheat cultivars, and sampling date were used (Table 1). Two water treatments were examined: rainfed and irrigation, under five N application levels: 0, 90, 180, 270, 360 kg hm$^{-2}$. Fifty percent of the N (urea) was applied pre-planting, while the remaining 50% was applied at the jointing stage. Base application of phosphate fertilizer (150 kg hm$^{-2}$ P$_2$O$_5$) and potash fertilizer (90 kg hm$^{-2}$ K$_2$O) was carried out in all plots before planting. Irrigation was carried out at the jointing stage (around mid-March) at a volume of 750 m$^3$ hm$^{-2}$. During the whole growth periods, the precipitation of Exp. 1–4 was 270.3, 235.9, 313.9, and 267.5 mm, respectively. The plots were arranged in a split-plot design with three replicates. The main plots were irrigation regimes and subplots were N rates. The plot size was 7 × 2.9 m in Zhengzhou city and 6 × 4.5 m in Shangshui city. The previous cultivated crop was maize. Mechanical deep tillage was used in both sites. Sowing and harvesting dates were 10.15–20 and 5.28–6.2, respectively. Other cultivation management (such as weed and pest control) measures were the same as in local high-yield fields.

**Table 1.** The experimental conditions, N fertilizer levels, irrigation regimes, and sampling date.

| Exp. No. | Season and Site | Cultivar | Soil Characteristics | Treatments | Sampling Dates (Days after Planting (DAP)) |
|---|---|---|---|---|---|
| Exp. 1 | 2015–2016 Zhengzhou | Zhoumai 22 | Type: fluvo-aquic soil, OM[①]: 17.26 kg$^{-1}$, Soil pH (CaCl2): 7.58, TN[②]: 1.04 g kg$^{-1}$, AN[③]: 111.37 mg kg$^{-1}$, AP[④]: 30.72 mg kg$^{-1}$, AK[⑤]: 128.36 mg kg$^{-1}$ | N rate (kg ha$^{-1}$): N0, N90, N180, N270, N360, Irrigation regime (m$^3$ hm$^{-2}$): rainfed and irrigation (750) | 23-Feb (RS), 1-Mar, 8-Mar, 15-Mar (JS), 22-Mar, 29-Mar, 5-Apr, 12-Apr, 19-Apr (AS), 28-Apr (131, 138, 145, 152, 159, 166, 173, 180, 187, 196) |
| Exp. 2 | 2016–2017 Zhengzhou | Zhoumai 27 | Type: fluvo-aquic soil, OM: 22.47 g kg$^{-1}$, Soil pH (CaCl2): 7.81, TN: 1.04 g kg$^{-1}$, AN: 142.32 mg kg$^{-1}$, AP: 75.90 mg kg$^{-1}$, AK: 152.44 mg kg$^{-1}$ | Same as Exp. 1 | 18-Feb (RS), 15-Mar (JS), 25-Mar, 6-Apr, 14-Apr, 23-Apr (AS), 30-Apr (128, 155, 165, 176, 184, 193, 200)_ |
| Exp. 3 | 2016–2017 Shangshui | Zhoumai 27 | Type: lime concretion black soil, OM: 21.33 g kg$^{-1}$, Soil pH (CaCl2): 7.25, TN: 1.36 g kg$^{-1}$, AN: 102.58 mg kg$^{-1}$, AP: 66.35 mg kg$^{-1}$, AK: 173.52 mg kg$^{-1}$ | Same as Exp. 1 | 19-Feb (RS), 14-Mar (JS), 28-Mar, 18-Apr, 22-Apr (AS), 28-Apr (124, 152, 166, 187, 191, 197) |
| Exp. 4 | 2017–2018 Shangshui | Zhoumai 27 | Type: lime concretion black soil, OM: 20.91 g kg$^{-1}$, Soil pH (CaCl2): 8.01, TN: 1.16 g kg$^{-1}$, AN: 135.45 mg kg$^{-1}$, AP: 40.21 mg kg$^{-1}$, AK: 150.27 mg kg$^{-1}$ | Same as Exp. 1 | 25-Feb (RS), 21-Mar (JS), 2-Apr, 14-Apr, 24-Apr (AS), 29-Apr (127, 151, 163, 175, 185, 190) |

OM: organic matter; TN: total nitrogen; AN: available nitrogen; AP: available phosphate; AK: available potassium; RS: reviving stage; JS: jointing stage; AS: anthesis stage. ①: Walkley–Black titration method; ②: Kjeldahl method; ③: alkaline diffusion method; ④: 0.05 M HCL; ⑤: 1M NH4OAc.

## 2.2. Analysis of N Content

Wheat was sampled from an area of 0.20 m$^2$ (0.5 m, two rows) in each plot. Plants were then separated into leaves, stems, and spikes, dried at 70 °C, then the weights of each component were obtained. The total N content of the leaves, stems, and whole plant (leaves + stems) was then determined after pulverization using the Kjeldahl method [26].

## 2.3. Data Analysis

The data for determination of critical N points were analyzed according to Justes et al. (1994) [11]. The tissue dry matter (leaf, stem, and plant), the corresponding N concentrations and wheat yield wheat yield were tested using an analysis of variance (ANOVA) with SPSS version 23.0. The means of each treatment were also compared using the least significant difference (LSD) test at a probability level of $p \le 0.05$. The regression lines were calculated separately for each group of data points using Microsoft Excel. The relative yield was obtained by dividing the yield at a given N rate by the highest yield among all N treatments.

## 2.4. Model Building

### 2.4.1. Establishment of a Dilution Curve Model for the Critical Nitrogen Concentration

The following formula was used to determine the critical nitrogen concentration dilution curve according to Justes et al. (1994) [11]:

$$Nc = aDM^{-b} \tag{1}$$

where Nc is the critical N concentration value (%), DM is the maximum dry matter accumulation (t hm$^{-2}$), a is the critical N concentration value when the dry matter is 1 t hm$^{-2}$, and b controls the slope of the curve.

### 2.4.2. Construction of Wheat Nitrogen Nutrition Index Models

The following formula was used to construct NNI models and evaluate the N nutrition status of the wheat plants [27]:

$$NNI = N_a/N_c \tag{2}$$

where NNI is the N nutrition index, $N_a$ is the measured N concentration (%), and $N_c$ is the critical N concentration (%).

### 2.4.3. Construction of Wheat N Deficient Models

The critical N accumulation (Equation (3)) and N deficiency models (Equation (4)) of wheat were subsequently derived using Equation (1). The derivation process is shown in Lemaire et al. (2008) [28]:

$$N_{cna} = 10aDM^{1-b} \tag{3}$$

$$N_{and} = N_{cna} - N_{na} \tag{4}$$

where $N_{cna}$ is the N accumulation under the critical N concentration (g m$^{-2}$), DM is the dry matter accumulation (t hm$^{-2}$), a and b are parameters, $N_{na}$ is the actual N accumulation under different treatments (g m$^{-2}$), and $N_{and}$ is the N deficit (g m$^{-2}$). An $N_{and}$ value greater than 0 represents insufficient N accumulation, while values less than 0 represent excessive N accumulation.

## 3. Results

### 3.1. Effects of Different Irrigation Conditions on Wheat Biomass

The effects of site, water regime, and N rate on wheat tissue dry matter are shown in Table 2. The results showed that these three factors and most of the interaction effects were significant in terms of tissue dry matter, highlighting the importance of the environmental conditions. Table 3 shows that increasing irrigation caused a significant increase in wheat growth, with a decrease in biomass under rainfed conditions with severe water limitations. The content of biomass increased with the increase of N rate, and N180, N270, and N360 are significantly higher than N0 and N90. The dataset from Exp. 2 was used as an example to show the wheat biomass at different growth stages (Figure 1). The stem and whole plant biomass showed a continuous increase. However, the biomass of the wheat leaves tended to increase then decrease with growth, and the maximum value appeared at 170−180 days after planting.

**Table 2.** The effects of site, N rate, and water regime on tissue dry matter.

| Source of Variation | df | Leaf Dry Matter (t hm$^{-2}$) | | Stem Dry Matter (t hm$^{-2}$) | | Plant Dry Matter (t hm$^{-2}$) | |
|---|---|---|---|---|---|---|---|
| | | Jointing | Anthesis | Jointing | Anthesis | Jointing | Anthesis |
| Site (S) | 1 | 2.692 | 5.033 * | 10.901 ** | 10.819 ** | 14.811 ** | 4.766 * |
| Water regime (W) | 1 | 4.895 * | 6.141 * | 10.548 ** | 44.992 ** | 18.093 ** | 49.878 ** |
| N rate (N) | 4 | 4.757 * | 4.684 * | 5.212 * | 25.228 ** | 17.064 ** | 33.142 ** |
| S*W | 1 | 4.926 * | 3.259 | 6.253 * | 3.209 | 8.854 * | 6.769 * |
| S*N | 4 | 2.327 | 1.886 | 3.764 * | 6.813 ** | 1.153 | 7.277 ** |
| W*N | 4 | 3.561 * | 3.445 * | 4.319 * | 4.255 * | 4.568 * | 5.031 * |
| S*W*N | 4 | 2.016 | 3.384 * | 3.357 * | 4.125 * | 1.854 | 6.011 ** |

\*\*, and \*, indicate that the F of the effect is significant for $p < 0.001$ and $0.001 < p < 0.05$, respectively.

Table 3. Effects of N rate and water on tissue dry matter under different years and sites in Exp. 2 and 3.

| Tissue Dry Matter (t hm$^{-2}$) | Water | DAP (Days) | Zhengzhou (2016–2017) | | | | | Shangshui (2016−2017) | | | | |
|---|---|---|---|---|---|---|---|---|---|---|---|---|
| | | | N0 | N90 | N180 | N270 | N360 | N0 | N90 | N180 | N270 | N360 |
| Leaf | Rained | 155$^Z$(152)$^S$ | 1.257e | 1.898d | 2.202c | 2.645b | 3.053a | 1.729d | 2.282c | 2.699b | 2.849a | 2.833a |
| | | 193(191) | 0.702e | 1.055d | 1.426c | 2.118b | 2.427a | 1.172d | 1.914c | 2.371b | 2.693a | 2.814a |
| | Irrigation | 155(152) | 1.909c | 2.101c | 2.615b | 2.931b | 3.276a | 1.683d | 2.162c | 2.658b | 2.867b | 3.317a |
| | | 193(191) | 1.251d | 1.811c | 2.333b | 2.462b | 2.815a | 1.501c | 1.923b | 2.441a | 2.643a | 2.785a |
| Stem | Rained | 155(152) | 1.229c | 1.648b | 1.931a | 1.944a | 1.521b | 2.079e | 2.987d | 3.159c | 3.473b | 3.886a |
| | | 193(191) | 4.240e | 5.138d | 6.556c | 7.734b | 8.176a | 4.687d | 5.428c | 6.371a | 6.426a | 6.027b |
| | Irrigation | 155(152) | 1.869b | 1.907b | 1.755b | 2.324a | 2.251a | 1.742e | 2.974c | 2.177d | 3.393b | 4.517a |
| | | 193(191) | 4.702e | 6.300d | 6.945c | 7.793b | 8.913a | 4.891d | 5.897c | 5.884c | 6.759b | 7.318a |
| Plant | Rained | 155(152) | 2.486e | 3.580d | 4.133c | 4.597b | 4.897a | 3.808c | 5.270b | 5.858b | 6.32a | 6.720a |
| | | 193(191) | 5.134e | 7.192d | 8.727c | 11.164b | 13.152a | 7.199c | 10.569b | 11.171ab | 11.811a | 11.508a |
| | Irrigation | 155(152) | 2.029e | 3.042d | 3.406c | 4.605b | 5.819a | 3.425d | 4.935c | 5.137c | 6.261b | 7.835a |
| | | 193(191) | 5.968e | 7.379d | 10.389c | 12.652b | 14.872a | 9.191e | 10.231d | 11.381c | 12.015b | 13.892a |

The small letters indicate significant difference at 0.05 level of probability among different N rates for each water regime level in each experiment by least significant difference (LSD) multiple comparison. Z: DAP of Zhengzhou; S: DAP of Shangshui.

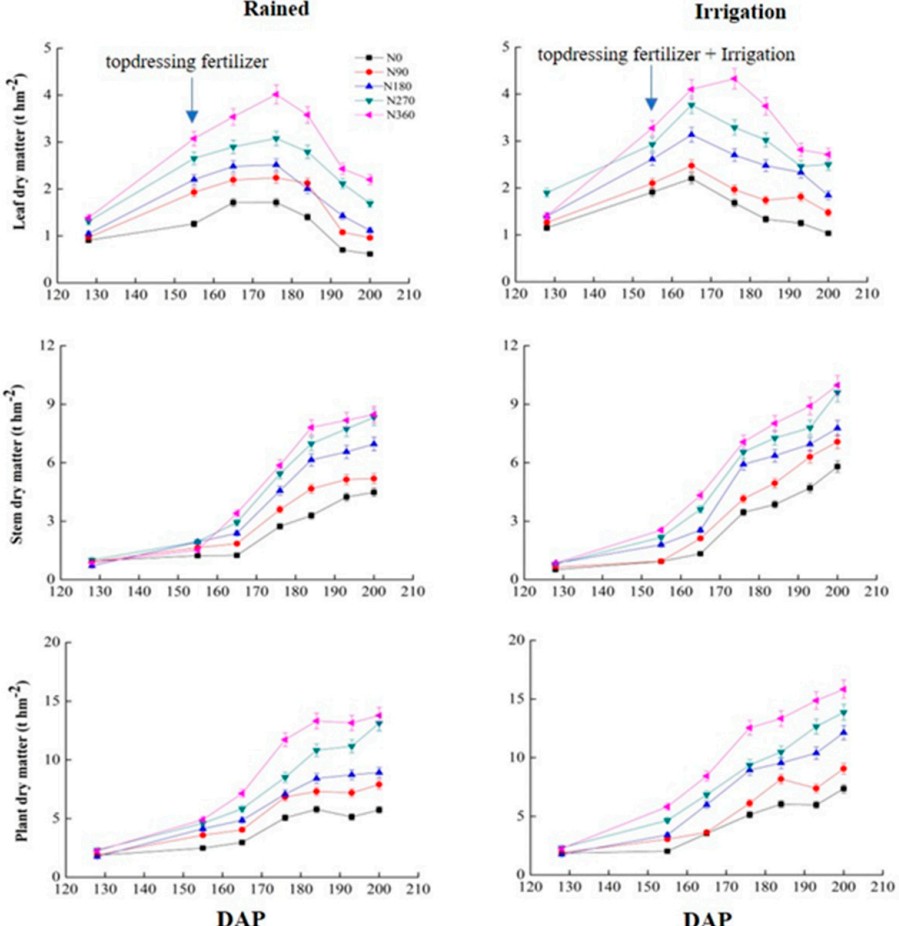

**Figure 1.** Effects of N application rates on wheat dry matter under different irrigation. DAP: days after planting. The datasets are from Exp. 2.

### 3.2. Effects of Different Irrigation Conditions on the N Content of Wheat

The effects of site, water regime, and N rate on wheat tissue N concentration are shown in Table 4. The results showed that these three factors and most of the interaction effects were significant in terms of tissue N concentration. In particular, the effect of N rate on tissue N concentration was extremely significant ($p < 0.001$). Table 5 shows that the tissue N concentration of irrigation treatment is higher than that of rained treatment. The effect of the N application rate on the N content of the leaves, stems, and whole plant was also consistent with the biomass results, increasing N application causing an increase in the N content. An increase in the water and N application rate therefore caused an increase in the N absorption capacity. The dataset from Exp. 2 was used as an example of the effects of different water and N levels on the N content of the leaves, stems, and whole plant at different growth stages (Figure 2). The N contents of the leaves, stems, and whole plant differed with growth. For example, the N content of the leaves showed an initial decrease followed by an increase then a second decrease, while that of the stems and the whole plant showed a continuous decrease.

**Table 4.** The effects of site, N rate, and water regime on tissue N concentration.

| Source of Variation | df | Leaf N Concentration (%) | | Stem N Concentration (%) | | Plant N Concentration (%) | |
|---|---|---|---|---|---|---|---|
| | | Jointing | Anthesis | Jointing | Anthesis | Jointing | Anthesis |
| Site (S) | 1 | 16.734 ** | 7.248 * | 30.832 ** | 5.618 * | 6.584 * | 5.721 * |
| Water regime (W) | 1 | 9.186 * | 9.891 * | 13.929 ** | 14.601 ** | 10.944 ** | 5.347 * |
| N rate (N) | 4 | 32.488 ** | 42.921 ** | 22.942 ** | 8.235 ** | 28.455 ** | 10.758 ** |
| S*W | 1 | 8.366 * | 3.172 | 11.660 ** | 5.071 * | 15.526 ** | 1.077 |
| S*N | 4 | 5.211 * | 6.105 ** | 8.618 ** | 3.053 * | 7.316 ** | 1.951 |
| W*N | 4 | 5.920 ** | 7.175 ** | 6.106 ** | 3.477 * | 6.8765 ** | 1.703 |
| S*W*N | 4 | 3.378 * | 8.082 ** | 3.025 * | 3.144 * | 5.029 * | 2.145 |

**, and *, indicate that the F of the effect is significant for $p < 0.001$ and $0.001 < p < 0.05$, respectively.

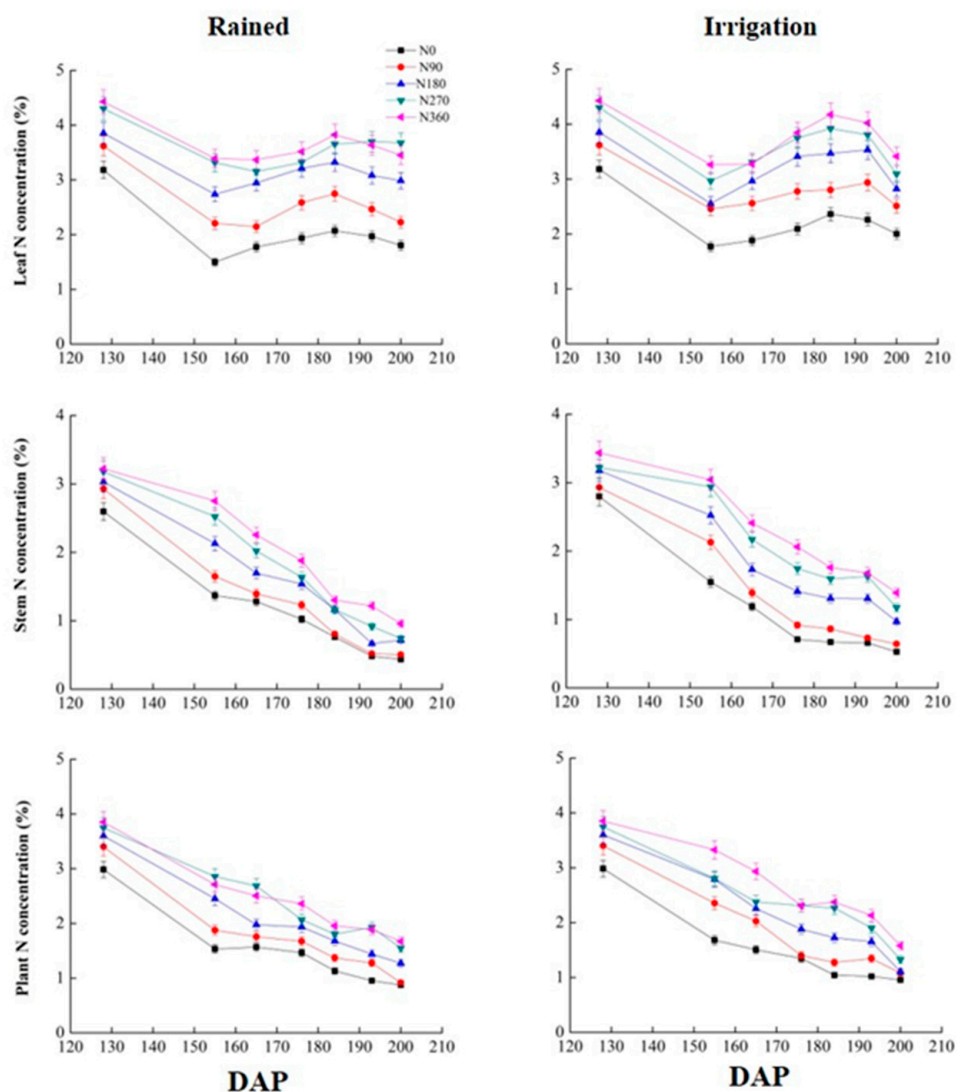

**Figure 2.** Effects of N application rates on the N concentration of wheat under different irrigation conditions. DAP: days after planting. The datasets are from Exp. 2.

**Table 5.** Effects of N rate and water on tissue N concentration under different years and sites in Exp. 2 and 3.

| Tissue N Concentration (%) | Water | DAP (Days) | Zhengzhou (2016–2017) | | | | | Shangshui (2016–2017) | | | | |
|---|---|---|---|---|---|---|---|---|---|---|---|---|
| | | | N0 | N90 | N180 | N270 | N360 | N0 | N90 | N180 | N270 | N360 |
| Leaf | Rained | 155(152) | 1.497d | 2.206c | 2.738b | 3.316a | 3.392a | 2.247d | 2.972c | 3.379b | 3.692a | 3.331b |
| | | 193(191) | 1.969d | 2.465c | 3.082b | 3.699a | 3.636a | 2.132d | 2.768c | 3.507b | 3.783ab | 3.843a |
| | Irrigation | 155(152) | 1.773d | 2.460c | 2.556c | 2.969b | 3.263a | 2.571d | 3.042c | 3.430b | 3.516b | 3.764a |
| | | 193(191) | 2.262e | 2.937d | 3.533c | 3.809b | 4.025a | 2.480c | 3.578b | 4.083a | 4.168a | 4.185a |
| Stem | Rained | 155(152) | 1.369e | 1.649d | 2.129c | 2.523b | 2.752a | 0.695b | 0.783b | 1.112a | 1.213a | 1.073a |
| | | 193(191) | 0.483c | 0.519c | 0.667c | 0.917b | 1.217a | 0.474b | 0.568b | 1.061a | 1.069a | 1.004a |
| | Irrigation | 155(152) | 1.548d | 2.127c | 2.521b | 2.940a | 3.041a | 0.726e | 1.156d | 1.433c | 2.136a | 1.844b |
| | | 193(191) | 0.658c | 0.729c | 1.305b | 1.628a | 1.678a | 0.641c | 1.072b | 1.148b | 1.227b | 1.482a |
| Plant | Rained | 155(152) | 1.534d | 1.877c | 2.453b | 2.856a | 2.712a | 1.935d | 2.603c | 3.053ab | 3.231a | 2.952b |
| | | 193(191) | 0.952c | 1.277b | 1.440b | 1.927a | 1.886a | 0.944c | 1.391b | 1.73a | 1.883a | 1.868a |
| | Irrigation | 155(152) | 1.682d | 2.357c | 2.789b | 2.803b | 3.328a | 2.106d | 2.965c | 3.129bc | 3.181b | 3.398a |
| | | 193(191) | 1.021e | 1.346d | 1.654c | 1.904b | 2.134a | 1.620d | 1.835c | 1.916bc | 2.073ab | 2.129a |

The small letters indicate significant difference at 0.05 level of probability among different N rates for each water regime level in each experiment by least significant difference (LSD) multiple comparison.

The irrigation conditions also had an obvious regulatory effect on the N content of the leaves, stems, and whole plant. Under rainfed conditions, the availability of soil N was limited, reducing N uptake and the transport capacity of the plants. Accordingly, the N content of the leaves, stems, and whole plant was low. In contrast, the N content of each component increased with increasing irrigation. Meanwhile, under rainfed conditions, N application had a relatively small effect on the N content, while under irrigation conditions, the effect of N fertilizer was relatively high.

### 3.3. Effects of Different Irrigation Conditions on the Dilution Model of Critical N Concentrations in Wheat

The critical N concentration dilution curves of the leaves, stems, and whole plant were subsequently obtained using Equation (1). As shown in Figures 3 and 4, the Nc model varied according to the test site, plant part, and irrigation conditions. Overall, the critical N concentration dilution curves of wheat in Zhengzhou and Shangshui under irrigation treatment were higher than those under rainfed treatment, suggesting that the critical N concentration is higher under irrigated conditions. The simulation accuracy was also higher under irrigation treatment than rainfed conditions. The critical dilution curves of the leaves, stems, and whole plant followed a similar trend, but parameters a and b differed. The range of a and b in Zhengzhou were 2.408–4.102 and 0.236–0.436, respectively, while in Shangshui they were 2.382–4.193 and 0.251–0.473, respectively. These differences may be related to differences in the climate and soil conditions between the two study sites. Moreover, the b values were higher under rainfed conditions in both sites, while the fitting accuracy was higher under irrigation than rainfed conditions. The slopes of all curves were also higher in Shangshui than in Zhengzhou, and the slopes of the stems and whole plant were higher than those of the leaves. Moreover, the slopes were all higher at an early compared to late growth stage. Overall, these findings suggest that the dilution of N concentrations was more pronounced in the stems and whole plant at early stages of growth.

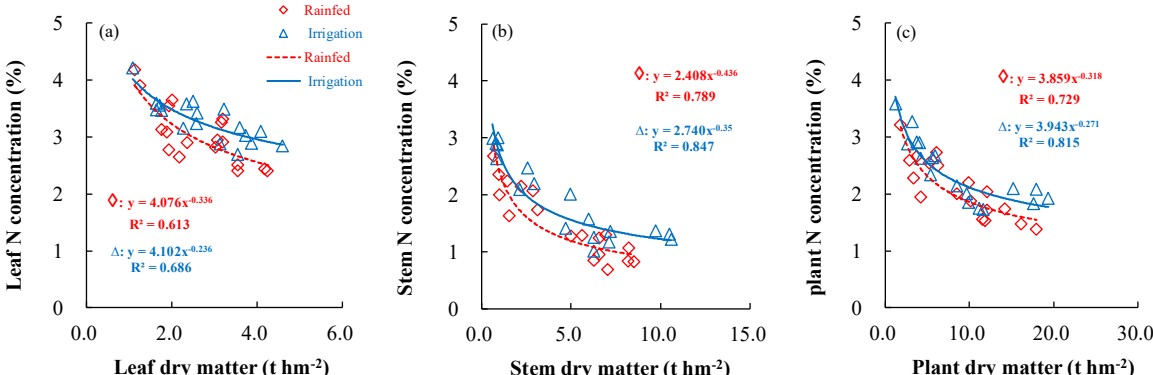

**Figure 3.** Critical N concentration dilution curves in wheat leaves, stems, and the whole plant under different irrigation conditions in Zhengzhou. The datasets are from Exp. 1 and 2.

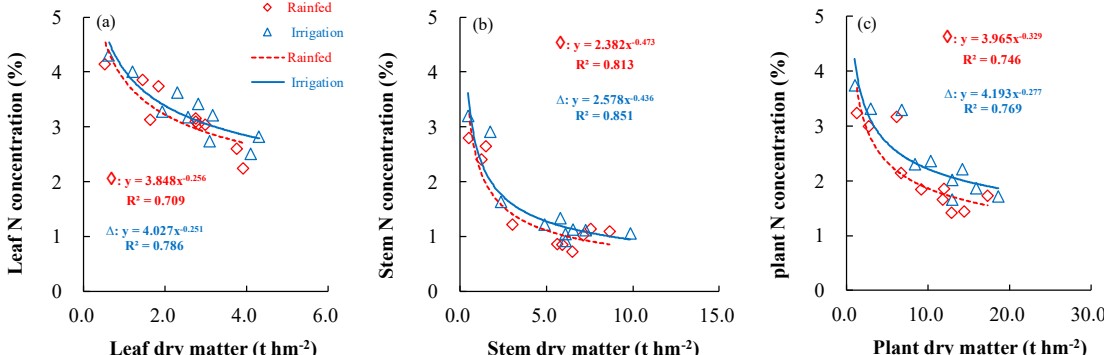

**Figure 4.** Critical N concentration dilution curves in wheat leaves, stems, and the whole plant under different irrigation conditions in Shangshui. The datasets are from Exp. 3 and 4.

Figure 5 compares the wheat plant critical N dilution curves obtained here with previously published curves. The Nc curve established by Ziadi et al. (2010) [12] ($y = 3.85x^{-0.57}$) in Canada was lowest, possibly because of the different varieties used (spring wheat rather than winter wheat). Meanwhile, the curve established by Justes et al. (1994) [11] ($y = 5.35x^{-0.44}$) in France had the highest critical N concentration in early stages of N dilution, but values were lower than those in this study in mid to late stages. These latter values obtained here were also higher than those in all related studies shown. The Nc curves established in Zhengzhou and Shangshui ($y = 3.943x^{-0.271}$ and $y = 4.193x^{-0.277}$, respectively) were also higher than the model established by Zhao et al. (2012) [13] ($y = 4.42x^{-0.44}$) in Eastern China and Yue et al. (2016) [14] ($y = 4.15x^{-0.38}$) in the North China Plain. At an early stage of N dilution, the Nc models in our study varied little from those in these two Chinese wheat growing areas; however, at later stages of growth, the differences between regions increased. Moreover, the Nc curves established by Zhao and Yue were closest, with a low critical N content compared to the high content in this study.

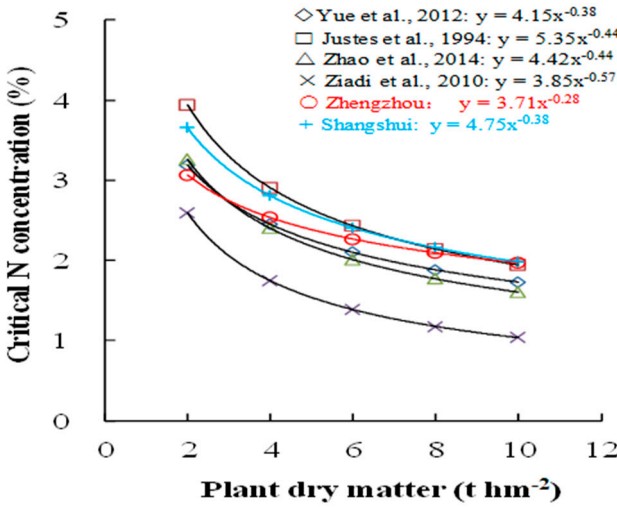

**Figure 5.** Comparisons of the critical N dilution curves of wheat plant obtained in Zhengzhou and Shangshui with previously published curves.

### 3.4. Effects of Different Irrigation Conditions on the N Nutrition Index of Wheat

The NNI can be used to effectively evaluate the N status of wheat plants. Figures 6 and 7 show the dynamic changes in the NNI of the leaves, stems, and whole plant under different irrigation and N application levels. Throughout growth, the NNI trends were the same under different irrigation

and N conditions, with an overall increase in NNI with increasing N application. However, values varied between the leaves, stems, and whole plant and there was a certain degree of fluctuation, ranging from 0.448–1.215, 0.568–1.207, and 0.561–1.261 in Zhengzhou and from 0.416–1.121, 0.393–1.217, and 0.438–1.131 in Shangshui, respectively. Under N0 treatment, the severe N deficiency resulted in NNI values well below 1 in the leaves, stems, and whole plant. Meanwhile, N90 and N180 treatment also resulted in values lower than 1, with growth limited by N deficiency. In contrast, N270 treatment resulted in NNI value of around 1, indicating a suitable N application rate, while N360 treatment resulted in values greater than 1, indicating sufficient or even excess N.

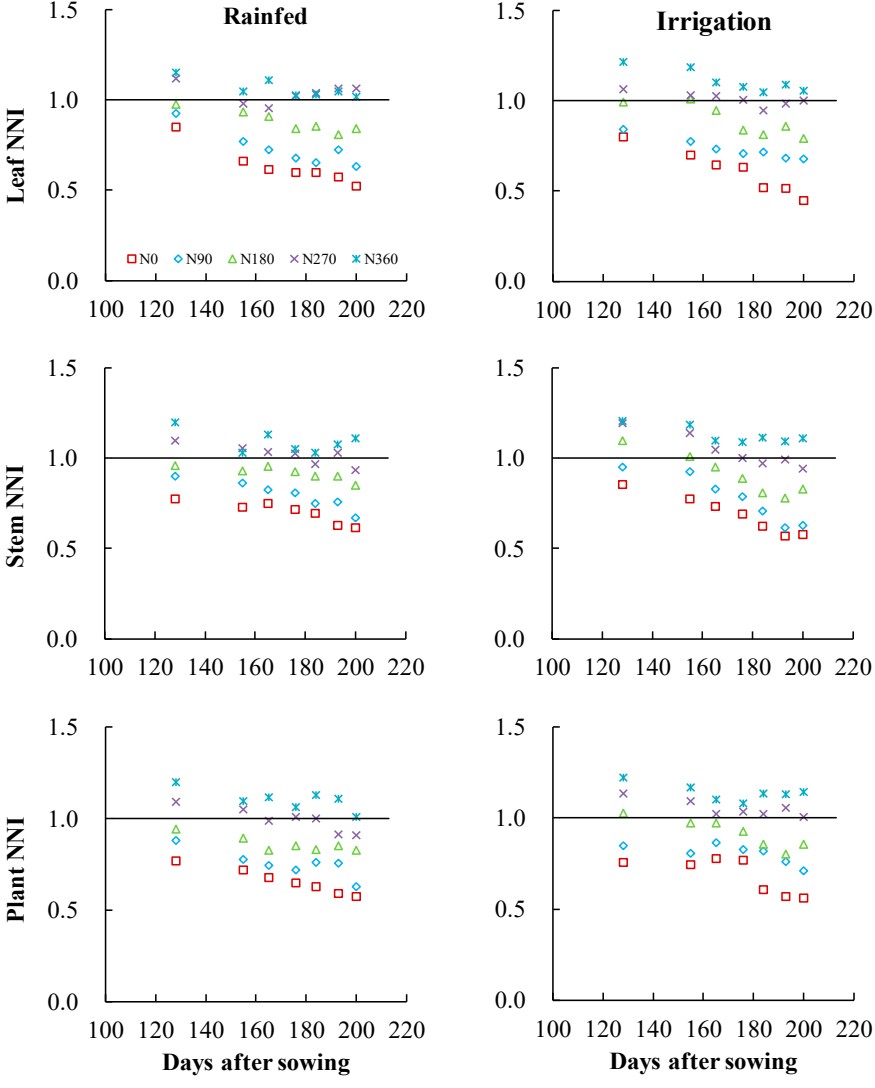

**Figure 6.** Effects of N application rates on the N nutrition index (NNI) of wheat under different irrigation conditions in Zhengzhou. The datasets are from Exp. 2.

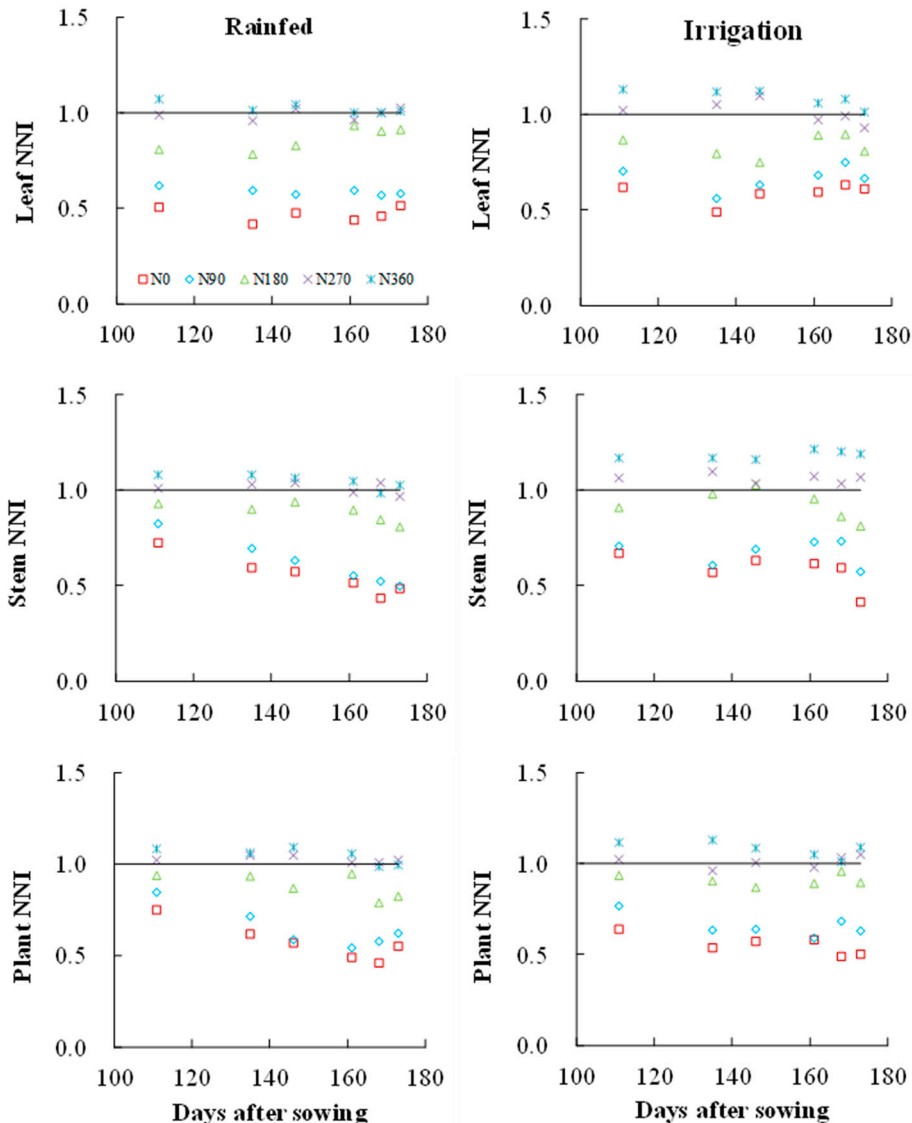

**Figure 7.** Effects of N application rates on the N nutrition index (NNI) of wheat under different irrigation conditions in Shangshui. The datasets are from Exp. 4.

### 3.5. Relationship between the NNI and N Deficiency and Relative Yield in Wheat

To further examine the applicability of the NNI in N management, the quantitative relationship between NNI and N deficiency was analyzed. As shown in Table 6, there was a significant negative correlation between NNI and N deficiency in the leaves, stems, and whole plant. In terms of fitting accuracy, the $R^2$ was higher under rainfed conditions than irrigation, and in the whole plant followed by the leaves and the stems. Overall, however, unified modeling was confirmed due to the small difference in the relationship between different test sites and different irrigation treatments. The response of NNI to changes in $N_{and}$ varied between the leaves, stems, and whole plant, with the slope expressed in the order of whole plant > leaves > stems. At an NNI of 0.5 to 1, rainfed and irrigated plants in Zhengzhou required 16.88 and 20.69 g m$^{-2}$ N, respectively, while in Shangshui only 13.894 and 16.688 g m$^{-2}$ N was required, respectively. These findings suggest that improving irrigation conditions requires an increase in N supply, with Zhengzhou requiring more N than Shangshui to meet crop growth needs.

**Table 6.** Relationship between the NNI and accumulative N deficit in wheat under different irrigation conditions.

| Location | Irrigation Condition | Component | Regression Equation | $R^2$ |
|---|---|---|---|---|
| Zhengzhou | Rainfed | Leaf | y = −27.81x + 28.24 | 0.808 |
| | | Stem | y = −26.87x + 28.63 | 0.708 |
| | | Plant | y = −27.25x + 28.46 | 0.812 |
| | Irrigation | Leaf | y = −32.75x + 33.33 | 0.732 |
| | | Stem | y = −32.40x + 32.94 | 0.679 |
| | | Plant | y = −36.32x + 37.57 | 0.752 |
| Shangshui | Rainfed | Leaf | y = −29.79x + 29.42 | 0.808 |
| | | Stem | y = −22.85x + 23.85 | 0.708 |
| | | Plant | y = −26.78x + 28.04 | 0.826 |
| | Irrigation | Leaf | y = −36.85x + 38.65 | 0.788 |
| | | Stem | y = −23.66x + 27.14 | 0.612 |
| | | Plant | y = −34.43x + 36.59 | 0.773 |
| All | | Leaf | y = −30.87x + 31.49 | 0.751 |
| | | Stem | y = −26.41x + 28.02 | 0.652 |
| | | Plant | y = −30.81x + 32.23 | 0.763 |

Note: the datasets of "All" are from Exp. 1–4.

The wheat yield was significantly affected by N rate and irrigation. As shown in Table 6, the yield of irrigation treatment was higher than that of rainfed treatment. The yield increased as the N rate increased from 0 to 270 kg hm$^{-2}$ but decreased at 360 kg hm$^{-2}$. The largest average yield of rainfed treatment in Zhengzhou and Shangshui was 7160.7 and 7235.4 kg hm$^{-2}$, respectively; it was 8075.2 and 7408.3 kg hm$^{-2}$ of irrigation treatment. The difference between N270 and N360 is not significant, but significantly higher than N0 and N90. Compared with the N0 treatment, the yield of N270 increased by an average of 131.7% (Z:2015−2016), 60.6% (Z:2016−2017), 83.5% (S:2016−2017), and 177.2% (S:2017−2018), respectively. It can be seen from Figure 8 that the leaves, stems, and whole plant NNI had significant correlations with relative yield, expressed by a quadratic curve, with R$^2$ of 0.771, 0.859, and 0.806, respectively. The relative yield increased with the increase of NNI. When NNI was near 1.0, the relative yield obtained a maximum value.

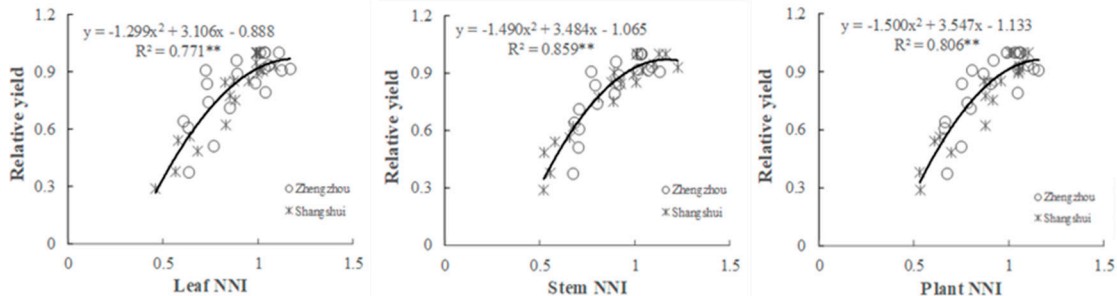

**Figure 8.** Relationship between the N nutrition index (NNI) and relative yield in winter wheat.

## 4. Discussion

N dilution occurs in crops as the dry matter content on the ground increases. In this study, we constructed critical N concentration models of wheat leaves, stems, and whole plants under rainfed and irrigation conditions. The resulting models were in line with the general model structure (N = aDW$^{-b}$) of critical N concentrations of C$_3$ and C$_4$ crops proposed by Greenwood et al. (1990) [3]. In the leaf and whole plant models, the a value was higher than in the stem model (Figures 3 and 4), possibly because N absorbed from the soil tends to be transported to the leaves to meet the needs of photosynthesis and respiration. As a result, the N concentration of the wheat leaves and whole plant will remain relatively high. Our Nc models of wheat plants obtained from irrigation plots in

Zhengzhou (y = 3.943x$^{-0.271}$) and Shangshui (y = 4.193x$^{-0.277}$) were higher than those constructed by Zhao et al. (2012) [13] (y = 4.42x$^{-0.44}$) and Yue et al. (2016) [14] (y = 4.15x$^{-0.38}$) for wheat grown in Eastern China and the North China Plain, respectively (Figure 5). This was possibly due to the climate and soil environment in our experimental area, which is more suitable for wheat growth, thereby increasing N needs. However, Our Nc models were lower than the model constructed by Justes et al. (1994) [11] in France (Nc = 5.35DW$^{-0.44}$) (Figure 5). This was possibly due to the high-protein variety used by Justes et al. (1994) [11] compared to the medium-protein variety used here. The roots of high-protein wheat are more developed and more active than those of medium to low-protein wheat. They also have higher N absorption, conversion, and utilization rates, and can absorb and assimilate more N during the growth process [29–31]. On the other hand, differences in the climate and soil environments of the test areas also affect N absorption and transformation.

In this study, we therefore constructed the critical dilution N concentration dilution curves of wheat under irrigation and rainfed conditions. As it is hard to control the ecological environment in the field, our level of model accuracy (0.613–0.851) was lower than that of the greenhouse crops. However, our curve models of the different plant components were consistent, despite differing model parameters. The parameters a and b, respectively, ranged from 2.408–4.102 and 0.236–0.436 in Zhengzhou and 2.382–4.193 and 0.251–0.473 in Shangshui under different cultivar, water availability, and N application rate. In addition, our findings revealed a notable decrease in the b value with increasing moisture (Figures 3 and 4). Overall, the Nc curves of the leaves, stems, and whole plant were higher under irrigation compared to rainfed treatment in both Zhengzhou and Shangshui, suggesting that the critical N concentration was higher with increasing water when the same biomass was obtained. This was possibly because wheat plants are weakened by water limitations, with a reduction in the N absorption capacity and subsequent decrease in the critical N concentration. In contrast, under adequate irrigation conditions, plants grow vigorously and N uptake capacity is enhanced, and as a result, the critical N concentration increases. Therefore, the synergistic effect of water and nitrogen conditions plays an important role in improving the nitrogen content of wheat. These findings also suggest that the N application rate should be lower under drought conditions, with a corresponding increase under improved water status. Under sufficient water conditions, the nutrients in the soil aqueous solution could be brought to the surface of the root with the flow of water. It provides favorable conditions for crops to obtain more nutrients. Having an in-depth understanding of the changes in the Nc curve of wheat under different water and N patterns and rational adjustment of water and fertilizer by agronomic measures are therefore important for helping protect the ecological environment and optimizing the efficiency of agricultural resources.

Effective N management ensures optimal N status for improving crop growth and grain yield. Inadequate N will limit crop growth, and N supplied in excess could cause environmental pollution [32,33]. In many crops, the NNI has been shown as an indicator of N status, and the NNI was significantly affected by N fertilizer level [10,34,35]. In this study, the NNI > 1 indicated excess N fertilizer and the yield reached saturation. The yield of N270 treatment in both Zhengzhou and Shangshui was the highest, with an NNI of around 1, suggesting that the optimal N application rate in this area is approximately 270 kg hm$^{-2}$ (Table 7, Figures 6 and 7). The previous cultivated crop in the test areas was maize, which requires a lot of fertilizer. The field of Shangshui is relatively weak in N fertilizer. The soil type of Zhengzhou is fluvo-aquic soil, and it is easy to leak water and fertilizer. Therefore, N270 kg hm$^{-2}$ is suitable in the test areas. There is a very significant power function relationship between NNI and relative yield of wheat (Figure 8). Liang et al. (2013) [35] also revealed that the relationship between NNI and relative yield in summer maize is a power function, and the determining coefficient reaches a very significant level. In addition, several studies have reported a close relationship between the NNI and N nutrient. Justes et al. (1994) [11] constructed a model describing the relationship between the NNI and N nutrient ratio in wheat, Moreover, Ata-Ul-Karim et al. (2013) [34] proposed a negative correlation between the NNI and N deficiency in rice varieties Wuxiangjing 14 and Lingxiangyou 18, with slopes of −217.56 and −316.53, respectively. Zhao et al.

(2012) [13] also revealed a negative correlation between the NNI and N deficiency in wheat varieties Yangmai 16 and Ningmai 13, with slopes of −186.29 and −152.81, respectively. In this study, we not only analyzed the relationship between whole plant NNI and N deficits, but also that between leaf and stem NNI and N deficits, providing more detailed models. However, the slopes in this study differed greatly from those in Zhao et al. (2012) [13], possibly due to differences in the varieties used and the cultivation conditions. As a result, NNI could be used to quantitatively evaluate the nitrogen nutrition status and help to improve the wheat yield through the real-time control of nitrogen fertilizer.

**Table 7.** Effects of N rate and irrigation regime on yield under different years and sites (kg hm$^{-2}$).

| Treatment | Zhengzhou | | Shangshui | |
| --- | --- | --- | --- | --- |
| | 2015–2016 | 2016–2017 | 2016–2017 | 2017–2018 |
| Rainfed | | | | |
| N0 | 2675.1 d | 4091.8 d | 3650.2 d | 2011.7 d |
| N90 | 5312.5 c | 5658.6 c | 5850.4 c | 3758.5 c |
| N180 | 6025.0 b | 6000.3 b | 6733.7b | 5900.2 b |
| N270 | 7175.1 a | 6750.3 a | 7542.1 a | 6928.7 a |
| N360 | 6578.0 ab | 6308.6 ab | 6842.1 b | 6343.6 ab |
| Irrigation | | | | |
| N0 | 4187.5 d | 5100.3 d | 4775.3 c | 3408.5 d |
| N90 | 5812.5 c | 7242.0 c | 5767.1 bc | 4041.8 c |
| N180 | 6475.1 b | 7650.3 b | 6525.4 ab | 6041.9 b |
| N270 | 8175.0 a | 7975.3 a | 7658.7 a | 7157.8 a |
| N360 | 7525.1 a | 7233.6 c | 6933.7 ab | 6654.1 a |

The small letters indicate significant difference at 0.05 level of probability among different N rates for each water regime level in each experiment by least significant difference (LSD) multiple comparison.

## 5. Conclusions

Real-time analysis of N nutrition status in wheat and precise regulation of N fertilizer use are extremely important for improving yield and N use efficiency. Based on a three-year field study, we constructed $N_c$, NNI, and $N_{and}$ models of wheat leaves, stems, and the whole plant under different ecological conditions and different water–N combinations. As a result, the $N_c$ and $N_{and}$ models constructed here were able to accurately reflect the N nutrition status N of wheat under different water and N conditions, thereby expanding on previous models created under conditions of adequate irrigation. However, despite these findings, our model was based on a 3-year field study only, and therefore requires further experimental analysis under different ecological and cultivation conditions. In doing so, we will be able to improve effective N diagnostics and estimation accuracy, while providing a universal model for N fertilizer application. In turn, this will improve the accuracy of wheat N nutrition analyses and improve the efficiency of N fertilizer management in different ecological environments.

**Author Contributions:** B.-B.G. and W.F. conceived the research concept; B.-B.G., X.-H.Z., M.-R.L., J.-Z.D. and Y.M. performed the experiments; B.-B.G. and W.F. wrote the paper; Y.-J.Z., L.H. and N.-Y.J. edited and revised the paper. All authors have read and agreed to the published version of the manuscript.

**Funding:** This work was supported by the National Natural Science Foundation of China [grant number 31671624], the Thirteenth Five-year Plan of National Key Research Project of China [grant number 2018YFD0300701], National Agriculture Technology Research System of China (CARS-03-01−22) and the startup funds from the Henan University of Science and Technology(13480087).

**Conflicts of Interest:** The authors declare no conflict of interest.

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
