# Peer review of "Establishment of Critical Nitrogen Concentration Models in Winter Wheat under Different Irrigation Levels"

_agronomy, doi:10.3390/agronomy10040556_

Round 1
Reviewer 1 Report
Thank you for revising the manuscript based on the previous comments. It has been greatly improved. Some questions and comments were annotated for clarification as follows.
1) Why was the effect of water regime on tissue dry matter at jointing stage extremely significant (Table 2) even though you applied irrigation only during (not before) the jointing period?
2) How about the interaction among site, year, water regime, and N rate (Table 2)?
3) Still I could not understand why you can say water efficient is greater than N effect. Your explanation was “the effect of N rate on tissue dry matter was significant, while the effect of water regime on tissue dry matter was extremely significant.” However, the p value shows only the significance of correlation, and you cannot use it for comparison.
Reviewer 2 Report
Dear authors,
I have reviewed the paper and your last corrections. I appreciate the effort you made to answer the reviewers’ comments, but I am still puzzled by the way you handled the data analysis in regard to the analysis of variance and to the post-hoc tests.
In Table 3 and 4 you presented results for dry matter and N concentrations, respectively. However, the anova table is reported only for dry matter (Table 2). Please add anova results for N concentration too.
Your experiments, as reported in Table 1, considers 2015-16 and 2016-17 in Zhengzou and 2016-17 and 2017-18 in Shangshoui. The design is unbalanced, so you should explain how you considered this nesting in the anova. Please also test and discuss the interaction term of water regime and N rate.
In tables 3, 4 and 6 it is not clear how you built the post-hoc comparisons. It seems to me that the different levels of the N treatment are compared within each water regime level within each experiment, but this should be stated more clearly. Please also report what kind of adjustment di you use for multiple comparisons.
Please also keep the header “sampling dates” (or “events”, or else) in Table 1: DAP is the measurement unit, but it has to be clear for the reader what kind of activity was performed.
Yours sincerely
Author Response
Please see the attachment.

This manuscript is a resubmission of an earlier submission. The following is a list of the peer review reports and author responses from that submission.
Round 1
Reviewer 1 Report
Major comments
The manuscript submitted by Bin-Bin Guo and co-authors deals with modeling of critical nitrogen (N) concentration of wheat plants under different sites, irrigation, and N fertilizer levels. The topic of this study is to figure out the N deficiency of wheat crop for N nutrition diagnosis in field condition, which we are lacking compared to the previous reports based on controlled condition, and it is within interest and scope of Agronomy. This manuscript is easy to read and has a lot of data based on 4 year-location experiments. Basically, I would like to see this paper published, but still I have serious concerns on several issues in this paper as follows.
1) Limitation of this research: Authors tried to link N deficiency to optimum N fertilization at last, and I agree with the importance of this attempt. However, the model of wheat N uptake regulation (section 3.6.) should be constructed and tested based on another experiment comparing the result of response of wheat N uptake to N application. Thus, the title should be modified not to include “optimization of N fertilizer use”, and the other parts regarding the model of N requirement in the manuscript should be deleted (e.g., L294-305, L391-395).
2) Rainfall and irrigation: There were no information of rainfall during the experiment in each site and year. Therefore, I could not understand how 750 m3 hm-2 (75 mm) of irrigation was critical for the wheat growth. Further, in Shangshui site where you used the same cultivar for two years, you can discuss more about significance of water availability for N uptake and critical N concentration. Overall, the interaction of rainfall and irrigation treatment was not clear.
3) Different cultivar in Zhengzhou site: Although authors noticed that Nc model can be influenced by the variety of wheat (L326-331), they did not mention about using the different cultivars in Zhengzhou site. Please add the reason why you changed the cultivar and explain how this difference affected in this study.
Minor Comments
Title:
As mentioned in major comments, please delete “optimization of N fertilizer use”.
Abstract:
- L22-23 This is not you found, is it? Please modify the sentence like; As previous researches reported, our results also showed that the critical N concentration and biomass followed a power function relationship.
- L26-28 You defined “N deficiency” as the difference between the N accumulation under critical N concentration and the actual N accumulation under different treatments (L150-152). If so, the increase of NNI with increasing N application does not necessarily reflect N deficiency.
- L30-33 As mentioned in major comments, the application of wheat N requirement model can not be discussed in this study. Please delete it.
Introduction:
- Can you use N for the abbreviation of nitrogen throughout the manuscript?
- L39 Wheat is an important crop not only in China but also all over the world. Maybe, you can emphasize your research strength here.
- L41 …sensitive to nitrogen level in soils.
- L53 “critical nitrogen concentration (of which part?)”
- L56 Please explain the abbreviations in the equation.
- L82 Soil moisture is an important...
- L105 NNI is not defined in the main text yet.
Materials and methods:
- L116 What does “the jointing stage” mean? Please clarify it using days after planting (DAP). Did you apply irrigation only one time or periodically during the experimental period?
- L118 Please clarify “other cultivation management measures.”
- What is the purpose of separating parts of wheat into leaf and stem (and whole plant)?
- L143-144 Please add the unit for Na and Nc.
- L151-152 Please add the unit for actual N accumulation and N deficit.
Results:
- I strongly recommend authors to make a new table summarizing the results of ANOVA of Exp.1-4 in 3.1 and 3.2 rather than showing the results of only Exp.2 in Figs. 1 and 2.
- L162-164 There is not enough information indicating that there were severe water limitations under rainfed conditions. Please show rainfall amount and fluctuation of soil water potential during the experimental periods. In addition, why can you say irrigation increased wheat growth? Please explain it based on statistical analysis data.
- L164-165 Again, why can you say water effect is greater than N effect? Please clarify it based on statistical analysis data.
- L166-169 This sentence should be moved to Discussion.
- L167 …with high nitrogen fertilizer application (in what context?), while…
- L191-194 These sentences should be moved to Discussion.
- L218 …the critical nitrogen dilution curves (of wheat? whole plant?) obtained…
- L224-225 Is the result of Nc in Zhengzhou derived from two different cultivars? It seems the accuracy of Nc curve in Zhengzhou is smaller than that in Shangshui. Have you considered to separate the Nc curve for different cultivars?
- L246-247 This sentence should be moved to Discussion. By the way, I doubt that NNI is a good diagnostic indicator of N nutritional status in wheat. Can you really say N is sufficient or deficient based on NNI under low water available condition? At least, you need more explanation to convince 1 is the good criteria for that.
- L264-268 These sentences should be moved to Discussion. Again, please show rainfall amount data during the experimental period to convince irrigation is important to increase N supply.
- L286-289 This sentence should be moved to Discussion.
- L294-305 This section should be deleted.
Discussion:
- L307-312 These sentences should be moved to Introduction.
- L314 Please explain how your models were in line with previous research.
- L315-317 The explanation of parameters can be deleted.
- L322 Are these equations in Zhengzhou and Shangshin obtained from the rain-fed plots? Please clarify it.
- L334-351 This paragraph should be moved to Introduction.
- L357 … in Shangshin under different cultivar, water availability, and N application rate.
- L365-367 This sentence is not clear. Please clarify the reason why N application rate should be lower under drought conditions.
- L371-374 These sentences should be moved to Introduction.
- L391-395 This part can be stated as a perspective for future research in Conclusion.
- L396-404 This paragraph should be moved to Conclusion.
Figure and Table:
Table 1
- Please add the rainfall data here.
- Please indicate sampling date using days after planting (DAP).
- Please indicate the abbreviation and analysis methods of soil characteristics.
Figure 1
- Please indicate date as DAP.
- Please indicate the date of irrigation and 1st and 2nd fertilizer application in figures.
- Apparently, the plant dry matter is larger than the sum of stem and leaf. Why? Please clarify it if you have additional procedure in Materials and Methods.
- Can you add any statistical information in this figure?
Figure 2
- Can you add any statistical information in this figure?
Figure 3&4
- Are these the combined data of two years in each site? If so, please indicate it in the title of figures, and add discussion regarding the effect of rainfall on leaf N in rainfed plot.
- Please use colored legends.
Figure 5
- Please use colored legends and highlight the lines of Zhengzhou and Shangshui.
Figure 6
- Is this combined data of Exp.1 and 2 using different cultivars? If so, please indicate in the title of figure.
Figure 7
- Is this combined data of Exp.3 and 4? If so, please indicate in the title.
Figure 8
- This figure could be deleted or used as a supplemental data. Instead, you can put additional data of whole data set in Table 2.
- Please use colored legends.
Table 2
- As suggested above, Figure 8 can be combined with Table 2.
Figure 9
- How did you calculate relative yield? Please clarify it in Materials and Methods.
- Please modify the legend in the right figure.
Table 3
- Please indicate the unit of yield and the meaning of alphabets.
Reviewer 2 Report
The paper aims at verifying the applicability of critical nitrogen concentration dilution curve and of nitrogen nutrition index of wheat to optimize N applications, considering rainfed and irrigated conditions. The authors used data on winter wheat from 4 experimental years to identify the N fertilization plans that caused the N content in wheat biomass to exceed or fall behind a critical threshold, in order to identify the optimal N application rate for their region.
Overall, the paper is accurate in representing its aims, the current state of research on this topic and the research areas where further improvement is needed. The paper is well structured, the amount of data collected is adequate, properly analyzed and interpreted and supported by comparisons with other studies on the same topic. The language is clear and easy to understand.
I find the paper of good quality and potentially suitable for publication after a minor revision.
Below are my comments and suggestions that I’d like the authors to answer:
Methodology
- Please provide more information about location, climate, weather, texture, and field management and agronomic practices (plot dimensions, previous cultivated crop, tillage, irrigation method, sowing and harvest dates).
- As for the data analysis, it is not clear what kind of post-hoc test was used (e.g. to get the letters of Table 3).
Results
- Please report a table with full results of your experiment and with results of the statistical analysis (anova or post-hoc results).
- In Figure 1, 2, 6, 7 the sampling dates are equally spaced but the number of days between dates is not constant. Please use 1 day as the unit for your x axis.
- Explain a little more in the figure captions which data points are represented (e.g. if the points are averaged between fertilization rate).
Discussion
- The optimal N rate found in this paper was 270 kg/ha. Is this fertilization rate optimal only for the same fertilization scheme used in the experiment (half as starter, half during jointing stage)? Is this the most common fertilization plan used in the region? Could the timing of fertilization influence the amount of N to apply? Could the precedent crop influence the N amount to apply? Please discuss these issues and, in general, circumstantiate a little more your fertilization advice (270 kg/ha can be a quite high value in many other regions of the world).
Best regards